# Episodes of fast crystal growth in pegmatites

Patrick R. Phelps [1✉], Cin-Ty A. Lee [1] & Douglas M. Morton[2]

Pegmatites are shallow, coarse-grained magmatic intrusions with crystals occasionally approaching meters in length. Compared to their plutonic hosts, pegmatites are thought to have cooled rapidly, suggesting that these large crystals must have grown fast. Growth rates and conditions, however, remain poorly constrained. Here we investigate quartz crystals and their trace element compositions from miarolitic cavities in the Stewart pegmatite in southern California, USA, to quantify crystal growth rates. Trace element concentrations deviate considerably from equilibrium and are best explained by kinetic effects associated with rapid crystal growth. Kinetic crystal growth theory is used to show that crystals accelerated from an initial growth rate of $10^{-6}$–$10^{-7}$ m s$^{-1}$ to $10^{-5}$–$10^{-4}$ m s$^{-1}$ (10-100 mm day$^{-1}$ to 1–10 m day$^{-1}$), indicating meter sized crystals could have formed within days, if these rates are sustained throughout pegmatite formation. The rapid growth rates require that quartz crystals grew from thin (micron scale) chemical boundary layers at the fluid-crystal interfaces. A strong advective component is required to sustain such thin boundary layers. Turbulent conditions (high Reynolds number) in these miarolitic cavities are shown to exist during crystallization, suggesting that volatile exsolution, crystallization, and cavity generation occur together.

[1] Department of Earth, Environmental and Planetary Sciences, Rice University, Houston, TX 77005, USA. [2] Department of Earth and Planetary Sciences, University of California, Riverside, CA 92521, USA. ✉email: prp2@rice.edu

Crystal size is traditionally linked to the cooling rate of a magmatic system[1–3]. In rapidly cooled systems (minutes to days), such as in in a lava that erupts and quenches, the resulting rock is characterized by fine (nm–μm) grain sizes. In rocks that cool more slowly (10–100 ky), such as in magmas that intrude or stall deep in the crust to form plutons instead of erupting to the surface, grains are coarser (up to cm size). These patterns seem straightforward: crystals take time to grow, so rapid cooling limits grain size, while slow cooling permits crystals to grow large. Granite is a silicic magma that cooled slowly and generated a coarse-grained rock. That same magma, when cooled quickly, becomes a fine-grained volcanic rock called rhyolite.

There are, however, observations that suggest the above understanding of grain growth in nature is not so simple. For example, it is not uncommon to find igneous rocks with feldspar crystals (e.g., megacrysts) several cm to even 10 cm in size in shallow plutons (e.g., porphyries), which, presumably, cooled faster than their finer-grained, deeper counterparts[4]. More strikingly, decimeter to even meter-sized crystals typify pegmatitic systems, but the small sizes of pegmatite bodies suggest the systems cooled quickly[5,6]. Many of the largest crystals on Earth thus appear to be found in seemingly short-lived systems[7]. Are crystals larger because they have had more time to grow, or do they represent anomalously rapid crystal growth? If the latter, what conditions promote rapid crystal growth, and might these conditions be more widespread than currently thought?

In many of these systems, water may play a crucial role in augmenting grain growth. For granitic pegmatites, water is key for reducing melt viscosities and crystallization temperatures while enhancing melt transport, allowing for much lower temperature crystallization to occur[8]. Here, we show that late-stage centimeter-scale crystals grew within hours. If such growth rates can be sustained, decimeter to meter-sized crystals in pegmatitic systems could have grown in days, with growth occurring in highly dynamic rather than static conditions. We suggest that the crystals grew from a free fluid phase within a turbulent convective system.

## Results

**The Stewart pegmatite.** To explore crystal growth rates in natural systems, we conducted a case study of quartz crystals from the Stewart pegmatite in the Pala district of southern California, USA. The pegmatites in this district are Cretaceous-aged and occur as dikes hosted in gabbroic plutons of the same age[9,10]. The pegmatites are thought to represent late-stage fluids fractionated from a hydrous gabbroic pluton that then form a tabular, compositionally zoned pegmatite body[9]. At its widest, the Stewart body is represented by a 50-m-wide dike with the outer layers composed of feldspar-rich granite (the upper and lower intermediate zones), a transitional perthite zone, and a coarse-grained internal core containing spodumene, lepidolite, heulandite, petalite, and albite (Fig. 1). Vertical structures (chimneys), which contain numerous miarolitic cavities, emanate from the internal core, penetrating upwards into the perthite zone. The chimneys host abundant course-grained tourmalines and albite, with tourmaline crystals in some cases approaching tens of centimeters in length. In these chimneys, tourmalines are set within a finer-grained matrix of albite, quartz, and lepidolite, with the base of the chimneys being lepidolite-rich. Quartz crystals in the chimneys, particularly within the miarolitic cavities, are often euhedral. The contacts between the chimneys and the pegmatite body are sharp, suggesting the chimneys represent late-stage fluids that diked into a mostly solidified upper zone of the pegmatite body. These chimneys make up ~50% by volume of the perthite zone and thus represent an important phase in the petrogenetic

evolution of the pegmatite[9]. Based on conductive cooling models, Webber et al.[6] suggest that the pegmatitic body cooled from 650 °C to below 550 °C within ~9 years after emplacement. This is an upper bound on cooling timescales (and lower bound on crystal growth rates) because any convective cooling, which was not considered by Webber et al.[6], would decrease estimated cooling timescales. Crystal growth rates inferred from pegmatite thermal lifespans would also be under-estimated because crystal growth occurs at different stages within the pegmatite's thermal evolution.

We investigated euhedral quartz crystals from the miarolitic cavities to constrain crystallization rates. The quartzes in these cavities are transparent, euhedral (doubly terminated), and range in size from a few cm to 10 cm. Their euhedral shapes indicate that they nucleated and crystallized from within the fluid making up the miarolitic cavities, not by nucleating along the walls. They are peculiar in that they have flattened crystal habits with the <1000> axis being short (the angled braces referring to the family of axes). Tourmaline appears to penetrate some quartz crystals perhaps as a result of quartz overgrowths.

**Quartz composition.** Quartz crystals were imaged at Rice University using cold cathodoluminescence (CL) microscopy, a type of visible wavelength luminescence induced by high-energy electrons emitted from a cold cathode[11]. Defects in the crystal lattice cause luminescence in some minerals. Since defects are often correlated with trace element substitutions, changes in CL color may indirectly indicate a change in trace element concentration[12]. Thus, changes in CL intensity were used to guide our geochemical study. The crystals were mounted in epoxy, cut, and polished. For CL imaging, an accelerating voltage of 12 kV was used with a vacuum current of 0.4–0.5 mA. Camera exposure was set to 4 s to maximize image quality. The resulting image for one of the crystals is shown in Fig. 2a (see Supplementary Fig. 1 for another CL quartz crystal).

Trace element compositions were quantified using a 213 nm New Wave laser ablation system coupled to a ThermoFinnigan Element 2 magnetic sector inductively coupled plasma mass spectrometry (ICP-MS) at Rice University. Laser spot analyses and line transects were made. Transects were performed using a 110 μm spot diameter with a ~13 J m$^{-2}$ fluence, a 5 Hz repetition rate, and a lateral laser velocity of 9 μm s$^{-1}$. The large spot size limits the spatial resolution of our transects to 100–200 μm. Analyses were done in medium mass resolution mode ($m/\Delta m = 3000$). The following masses were measured: Li$^7$, Na$^{23}$, Mg$^{25}$, Al$^{27}$, Si$^{30}$, P$^{31}$, K$^{39}$, Ti$^{49}$, Zn$^{66}$, Ga$^{69}$, Ge$^{73}$, and Ge$^{74}$. Signals were background-corrected, normalized to Si$^{30}$, and converted to concentrations using synthetic glass NIST 612[13] and two natural rhyolitic obsidians M3–33 and M3–86[14] as external standards and forcing the sum of all measured metals (in oxide form) to equal 100%.

Three quartz crystals were mapped using CL. The CL of one crystal is shown in Fig. 2a and is representative of the color zones seen in CL from the pegmatite. In CL, the quartzes consist of three sharply demarcated zones: a white core zone, an orange middle zone, and a purple outer rim. The white core consists of two subtle internal zones: a dimmer inner zone surrounded by a thin, slightly brighter mantle. We consider these two zones to be internal parts of the core zone because their CL colors are only subtly different. Throughout the grain, transient blue luminescence mainly in the orange and purple zones, which may be related to electron beam-induced rearrangements of alkali trace elements at the nanoscale[15,16], was observed.

A representative compositional transect (A–A′) across all three CL zones is shown in Fig. 2b (see Supplementary Figs. 2–7 for

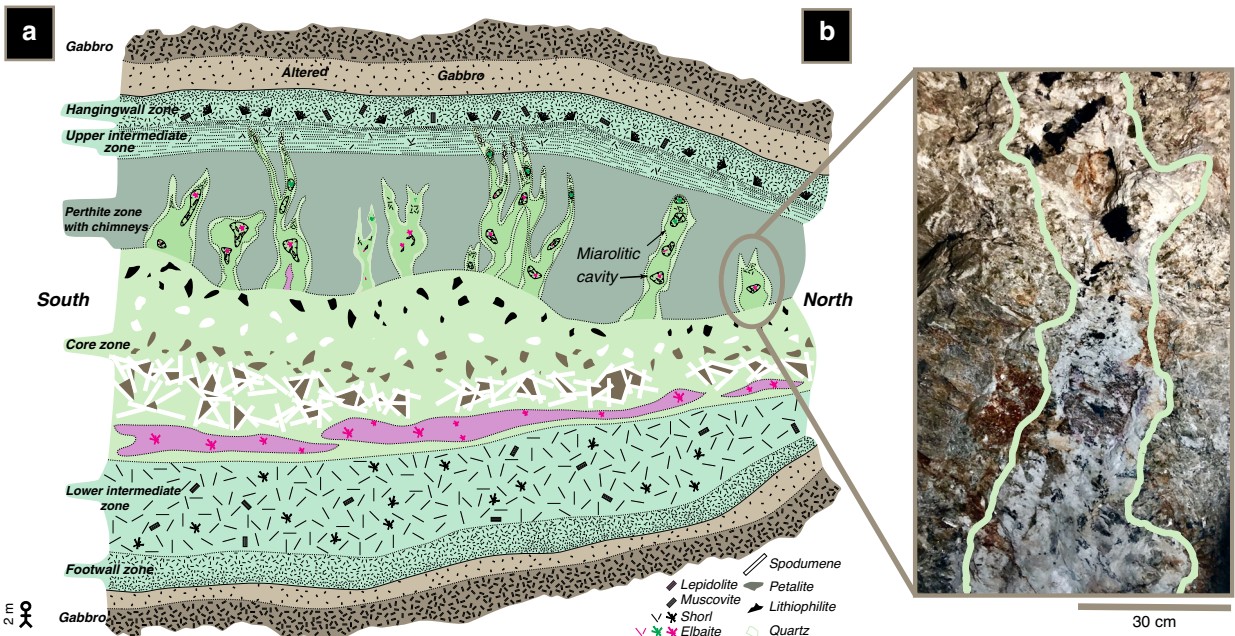

**Fig. 1 Cartoon of the Stewart pegmatite with an image of a miarolitic cavity. a** Schematic diagram of the Stewart pegmatite modified from Morton et al.[9]. A 2 m tall person for scale on the lower left. Outer and intermediate zones are granitic in composition. **b** An example of a small miarolitic cavity (chimney) that propagated from the core zone into the solidified upper zone. The miarolitic cavity contains tourmaline and quartz crystals set within a fine-grained matrix of lepidolite. Lighter margins of the cavity are defined by white albite.

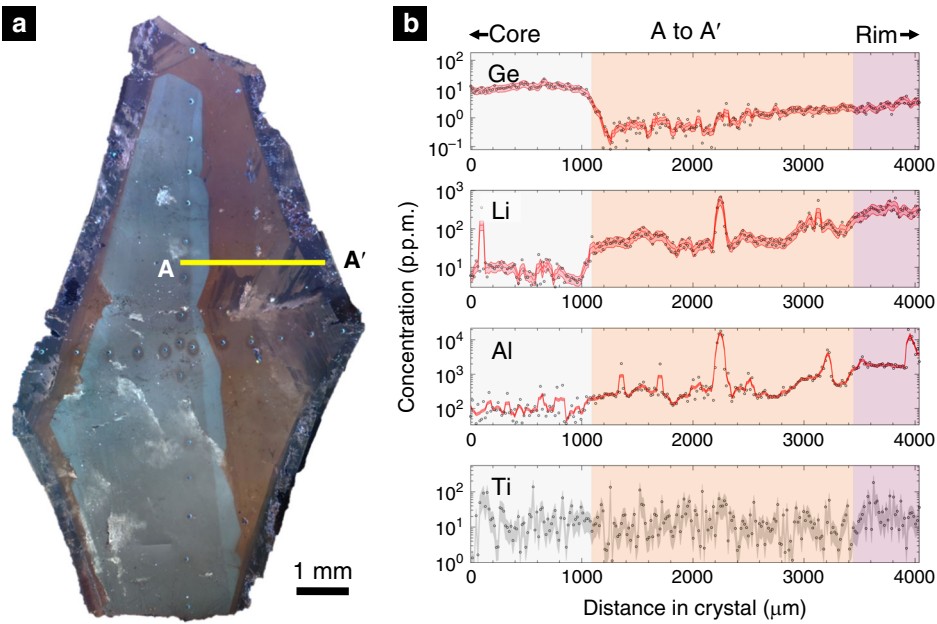

**Fig. 2 Cathodoluminescence image of a quartz crystal and corresponding trace element transect. a** Cathodoluminescence (CL) image of a quartz crystal from the Stewart pegmatite. Yellow line represents trace of geochemical transect. The plane of the crystal is part of {1000}. **b** Geochemical transect A–A′ for Ge, Li, Al, and Ti in p.p.m. by weight. Dots represent data and red lines represent moving averages. Red envelopes represent standard error of the mean based on $3\sigma$ of point analyses and a three-point moving average. Gray envelope in Ti represents 1 standard deviation to the data points. The error values are 26%, 28%, 8.5%, and 57% for Ge, Li, Al, and Ti, respectively. Background colors correspond with CL colors in **a**. Ti is scattered due to measurements being near detection limits.

other transects). Transitions in CL color or intensity correlate with broad changes in trace element concentrations[12]. In the white core region, most elements show minimal variation with distance (no compositional change occurs across the internal CL transition within the white core zone discussed above). For example, in the white core zone, Li and Al are relatively constant, while Ge increases outwards from 10 to ~20 p.p.m. before

decreasing back to 10 p.p.m. At the contact between the white core and orange middle zone, marked decreases in Ge and increases in Al and Li are observed. In the orange middle zone, Ge, Al, and Li show gradual outward increase in concentrations, punctuated by spikes in Al and Li due to inclusions (confirmed optically but of unknown mineralogy due to their small size). At the contact between the orange middle and purple outer zone, Al

and Li increase markedly, but no sharp changes are seen in Ge. In the purple outer zone, Al and Li flatten quickly and even decrease slightly, while Ge continues to increase.

Ti concentrations within the core zone of the crystal are near detection limits and are to within error constant between 10 and 15 p.p.m. We can use these Ti contents to obtain a maximum bound on temperature using Ti in quartz thermometry. Assuming $TiO_2$ activity of 0.6 (no Ti-bearing phases) and 2 kbar, we arrive at ~600 °C using the thermometer of Wark and Watson[17]. The thermometers of Huang and Audétat[18] and Thomas et al.[19] allow for pressure dependence, lowering the temperature to between 460[19] and 540 °C[18]. Adopting a lower $TiO_2$ activity would yield higher temperatures, but if $TiO_2$ contents are high due to rapid crystal growth (see below), lower temperatures would be estimated. While we cannot derive an exact temperature for the formation of these quartzes, it would seem that the quartzes formed at temperatures no >540 °C.

**Kinetic versus external control on compositional zoning.** We interpret the CL transitions as reflecting original differences in the density of defects or trace element impurities associated with sudden changes in the crystal growth rate. The broader Li, Ge, and Al concentration profiles across the sharp CL zones are larger than the spatial resolution of our laser spot size and may suggest diffusive relaxation of these elements. However, the width of each of these concentration profiles is similar, which is reflected in the strong coupling of Li and Al concentrations (Fig. 3). The similarity in the length scales of these transitions, if related to diffusion, is unexpected because these elements have markedly different diffusivities. For example, at temperatures of ~600 °C, tracer diffusivities in quartz are $5 \times 10^{-12}$ m² s⁻¹ (following Verhoogen[20]) for Li, $3 \times 10^{-23}$ m² s⁻¹ for Al (following Tailby et al.[21]), and ~$1 \times 10^{-32}$ m² s⁻¹ for Ge (assuming Ge behaves similarly to Si self-diffusion[22]). In particular, the extremely low diffusivity of Al would require nearly 2 My to generate the Al profiles by diffusion assuming an initial step function profile—far longer than the lifespan of the pegmatite. No diffusive relaxation is thus predicted for Ge and Al.

The presumably high diffusivity of Li, however, should predict measurable diffusive relaxation. For example, over the lifespan of the pegmatite (~9 years), Li should have diffused over a length scale of 7.5 cm—larger than the crystal sizes in this study. However, the fact that Li correlates strongly with Al suggests that

$Li^+$ in quartz is charge-coupled with $Al^{3+}$, suggesting that published Li tracer diffusivities may not be applicable. Instead, Li might only diffuse as fast as Al due to charge coupling. The possibility that Li diffuses very slowly is at odds with some studies that have inferred short residence times of crystals in magmatic systems from Li concentration profiles[23]. Our findings are similar to the observation of very slow diffusion of Li in zircons due to coupling with slow-diffusing rare-earth elements[24]. The strong coupling of Li to slowly diffusing Al and Ge inferred here suggests that there may have been limited subsolidus diffusive relaxation of Li, Al, and Ge across these boundaries.

The origin of compositional differences between and within each CL zone are thus of interest. Possibilities include changes in fluid composition (external forcing) or changes in the nature of crystal growth (internal processes). For the following reasons, we do not favor changes in fluid composition as the primary driver. Although quartz Ti concentrations (varying from 10 to >50 p.p.m., depending on location within the crystals) in this study are similar to those in quartz from granitic plutons, Al concentrations (>1000 p.p.m.) are significantly higher than those in quartz from granitic plutons and most pegmatites (Fig. 4). Given that Al is highly incompatible in quartz[25], these remarkably high Al concentrations would require fluids or melts with unreasonably high Al contents. We also note that the trace element composition of magmas or fluids undergoing differentiation should evolve in predictable ways: progressive crystallization leads to enrichment of incompatible elements and depletion of compatible elements. Li and Al are incompatible in quartz[25,26]. Ge is similar to Si[27–29] and should thus be compatible in quartz, as confirmed by He et al.[30]. Li and Al should thus be systematically anticorrelated with Ge, if their concentrations are controlled by a fluid or magma undergoing crystal fractionation. Instead, the relationship of Li and Al with Ge across the entire quartz grain shows more complexity.

While the possibility of external changes in fluid composition cannot be completely excluded, adopting such a hypothesis would not only require highly unusual fluid compositions, but very large and sudden changes in fluid composition. For these reasons, we also explore the possibility that the trace element compositions reflect kinetic processes, specifically the effects of rapid crystal growth.

**Discussion**

In this section, we explore how changes in crystal growth rate can affect how trace elements are incorporated into crystals[31–33]. A trace element will partition into a mineral according to its

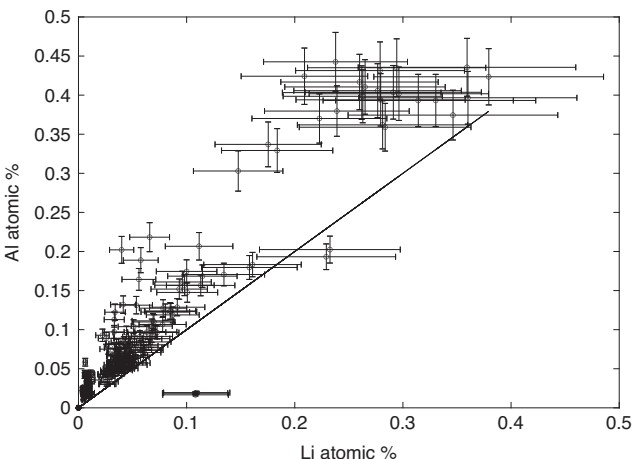

**Fig. 3 Al versus Li in quartz.** Line represents 1:1 correspondence of Li and Al. Correlation of Li with Al indicates possible charge coupling. Other monovalent cations, such as Na, may account for the remaining Al. Error bars are 3σ based on internal measurement precision.

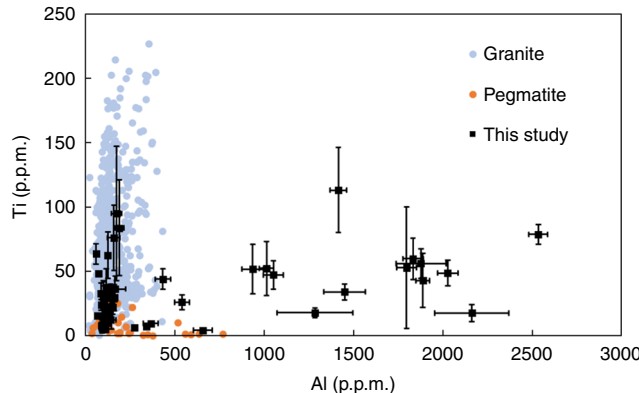

**Fig. 4 Ti versus Al concentrations in quartz from granites and pegmatites.** Granitic data compiled from Ackerson et al.[60] and pegmatite data from Garate-Olave et al.[61] and Götze et al.[15]. Error bars represent 3σ. Data for this study are from point analyses.

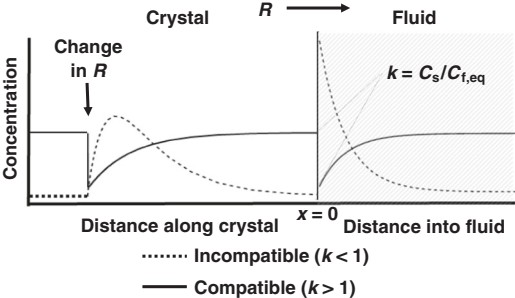

**Fig. 5 Trace element response due to changes in growth rate.** Shown is a conceptual diagram of how crystal growth rate ($R$) affects trace element distributions associated with a sudden change in growth rate. The figure shows the transient case in which growth rate suddenly increases from a previous steady state. The concentration ($C$) in the fluid near the crystal rises or falls, depending on the element's compatibility ($k$, the equilibrium ratio of the solid concentration to the liquid concentration), which is recorded in the crystal as an enrichment or depletion in the crystal, respectively. If the growth rate stays constant, the fluid concentration will eventually reach a new steady state, which will be reflected in a constant $C$ in the crystal as a function of distance, as distance represents time. The concentration profile in the fluid at steady state will represent a static chemical boundary layer.

equilibrium partition coefficient ($k$), which is the concentration in the crystal divided by the concentration in the growth medium (fluid or liquid) immediately adjacent to the growing crystal surface at equilibrium ($k = C_s/C_l$). However, if the crystal grows at rates comparable to or faster than the element can diffuse in the fluid, a chemical boundary layer in the fluid develops near the advancing crystal-fluid interface (Fig. 5). For an incompatible element ($k < 1$), crystal growth leads to rejection of the element from the crystal and subsequent enrichment of the element in the fluid boundary layer. The reverse is true for compatible elements ($k > 1$), wherein preferential incorporation of the element into the crystal leads to a depleted chemical boundary layer in the fluid. Qualitatively, the width of the boundary layer is controlled by the relative rates of crystal growth and diffusive or advective homogenization in the fluid. For example, fast crystal growth compared to diffusion in the fluid will result in extremely enriched and narrow chemical boundary layers for incompatible elements (and extremely depleted and wide boundary layers for compatibles). If diffusive timescales in the crystal are longer than crystal growth timescales, crystal zoning will reflect changes in the composition of the boundary layer, not the composition of the far-field fluid or magmatic growth medium.

These concepts can be placed in a quantitative framework following an advection-diffusion model with the crystal–liquid interface as a moving reference frame[31,33–35]

$$\frac{\partial C_l}{\partial t} = D\frac{\partial^2 C_l}{\partial x^2} + R\frac{\partial C_l}{\partial x}, \qquad (1)$$

where $C_l$ is the concentration of the trace element in the liquid, $D$ is the diffusivity of that element in the liquid, $R$ is the growth rate, $t$ is time, and $x$ is the distance from the growth interface in the liquid. The boundary conditions are:

$$D\frac{\partial C_l}{\partial x}\Big|_{x=0} = R(k-1)C_l|_{x=0} \quad \text{at } x = 0, \qquad (2a)$$

$$C_l = C_{eq} \quad \text{as } x \to \infty, \qquad (2b)$$

where $C_{eq}$ is the concentration in the far-field fluid. The interface boundary condition (2a) assumes flux continuity (Ch. 9[35]). The diffusive flux of an element out of the fluid and into the crystal is governed by Fick's law, $J_{diff} = -D\frac{\partial C_l}{\partial x}$. This flux is balanced by the

growth flux, which is given by $J_{rej} = (C_l - C_s)R$ or $J_{rej} = (1 - k)C_l R$. The boundary condition at infinity (2b) assumes that the fluid composition is constant at distances greater than the diffusive length scale within the fluid. A fundamental assumption in this approach is that reaction kinetics at the crystal surface are fast compared to diffusion and can be neglected. If this is not the case, a modified partition coefficient, like that detailed in ref. [36], must be included. Because there are no constraints on rate dependent $k$ for our system, we assume $k$ to be constant from here on.

By setting the time derivative to zero in Eq. 1, we can explore the steady-state solution for the composition in the liquid[31]:

$$C_l = C_{eq}\left(1 + \frac{1-k}{k}\exp\left(-\frac{Rx}{D}\right)\right). \qquad (3)$$

Equation 3 gives the form of the steady-state concentration profile in the fluid relative to the far-field fluid composition $C_{eq}$. The maximum deviation from the far-field concentration $C_{eq}$ occurs at the interface and scales with $1/k$, as seen by setting $x = 0$ in Eq. 3. The deviation (positive for $k < 1$, negative for $k > 1$) of the liquid composition from equilibrium decreases exponentially with distance away from the growing crystal interface.

Importantly, *at steady state*, the concentration in the solid $C_s$ remains constant as the crystal grows and is equal to the product of the partition coefficient $k$ and the concentration in the liquid at $x = 0$, that is, $k \times (C_{eq}/k) = C_{eq}$. This holds for a static growth medium. If the growth fluid flows, then a different steady state must be considered.

The concentration in the crystal at the onset of growth is $C_s = kC_{eq}$ and gradually evolves to the steady solution given by Burton et al.[32],

$$C_s = C_{eq}\left(\frac{k}{k + (1-k)\exp\left(-\frac{R\delta}{D}\right)}\right), \qquad (4)$$

where $\delta$ is the steady-state width of the chemical boundary layer in the fluid, and $C_s$ is the composition of the crystal at the interface. This is found by restating Eq. 2b with $x \to \delta$. The quantity $R\delta/D$ is a Peclet number representing the relative strengths of advection versus diffusion. If the Peclet number is small, the composition of the crystal approaches equilibrium with the far-field fluid, that is, $C_s = kC_{eq}$. For very large Peclet numbers, such as for large growth rates, the crystal composition $C_s$ approaches the far-field liquid composition $C_{eq}$[35]. In a static fluid, the steady-state width of the chemical boundary layer $\delta$ scales with $D/R$ and represents the distance over which the deviation from equilibrium with the far-field fluid decays by $e$-fold, as shown by Eq. 3. Thus, for constant diffusivity $D$ in the fluid, an increase in the crystal growth rate $R$ will translate to a smaller $\delta$. If, however, the fluid itself is advecting, the chemical boundary layer can become physically thinned, such that $\delta$ becomes a complex function of $D$ and fluid flow rate. A steady state can still exist, but $\delta$ is thinner due to advective erosion of the boundary layer in the fluid[33].

The time-dependent solution of Eq. 1 in a static fluid can be derived for the case in which the system is initially at steady state under a constant initial growth rate $R_i$ (Eq. 3), but then experiences an instantaneous change to a new and constant growth rate $R_f$[34]. How the system returns to steady state after a step change in $R$ is given by Eq. 5 (see "Methods"). Qualitative solutions are shown in Fig. 5 to give examples of the behavior of this solution. The transient solution reveals that upon a sudden increase in growth rate, incompatible elements in the growing crystal will first increase before falling back to a steady state, whereas compatible elements first decrease before rising back to steady state. This behavior relates to the time required for $\delta$ to respond, translating to the peaks in concentration for the incompatible case and valleys for the compatible one. The primary controls on

the behavior of the response are initial growth rate $R_i$, which controls initial steady-state concentration and hence the magnitude of the response, and the ratio of $R_i/R_f$ and diffusivity, which together control the shape of the response after an increase in growth rate.

We first explore compositional variations in the quartz in the context of rapid but steady-state crystal growth. Exploring steady state gives insight into the direction systems should evolve towards. Equation 4 can be evaluated for different Peclet numbers. Doing so requires constraints on the mineral/fluid partition coefficient $k$, the diffusivity $D$ of the element in the fluid, and the concentration of the far-field fluid $C_{eq}$. To better constrain results, we use two elements: Al and Ge. For Al, the mineral/fluid partition coefficient has not been measured, so we use published mineral/melt partition coefficients of ~0.02[25]. For Ge, following the approach of Hofmann et al.[37], we can use the fact that Ge/$SiO_2$ ratio changes only slightly over a range of $SiO_2$ concentrations from andesitic to rhyolitic compositions (Supplementary Fig. 8) to infer that Ge partitioning into quartz is similar to that of Si[30]. Therefore, $k_{Ge}$ can be estimated by constraining how Si partitions into quartz relative to a fluid, which can be constrained from Si solubility in the fluid. Because the quartz precipitated in miarolitic cavities, a water-dominated fluid is assumed rather than a hydrous silicate melt as the growth medium[8,9,38,39]. At 540 °C and 150–200 MPa[6,10] (5–7 km emplacement depth) and using the Si solubility data of Fournier and Potter[40] and Burnham et al.[41], we arrive at a $k_{Si}$ of ~200. We assume that $k_{Ge}$ for quartz is of similar magnitude. Recent work on natural quartz–silicate melt partitioning by He et al.[30] confirms that Ge is compatible in quartz and only slightly less compatible than Si.

Diffusivity of Al and Ge in the fluid is determined following methods outlined in Nigrini[42] and Yuan-Hui and Gregory[43]. Their parameterizations are applicable only to +1 to +3 valence states. The likely valence of Ge is +4, but because $Ge^{4+}$ has a similar cation size to $Al^{3+}$ of 0.5 pm[27], we approximate Ge diffusivity with that of $Al^{3+}$ for the above temperatures of interest and allow for an uncertainty of two orders of magnitude (~$7 \times 10^{-9 \pm 1}$ $m^2$ $s^{-1}$). This uncertainty in diffusivity more than accounts for the uncertainty in temperature. Changes by ±100 °C do not significantly affect the diffusivities (<15% changes).

Finally, $C_{eq}$ can be estimated by dividing the concentration at the center of the crystal (~15 p.p.m. for Ge and ~100 p.p.m. for Al) by $k$ as inferred above. This approach is justified because the trace element concentration of a crystal at the onset of crystallization (core) will initially be in equilibrium with the far-field fluid, $C_s = kC_{eq}$, because no boundary layer in the fluid should have yet developed.

In Fig. 6, we compare Ge and Al concentrations to modeled steady-state values of Ge and Al for different values of $R\delta$ (Eq. 4), that is, the product of crystal growth rate and boundary layer thickness. We have assumed constant diffusivity so that variations in $R\delta$ effectively represent changes in Peclet number. At steady state, $\delta$ is expected to be a constant for a given crystal growth rate and diffusivity (and fluid advection), but because we are considering changes in $R$, we do not a priori know what $\delta$ is, so $R\delta$ is varied. In any case, increases in $R\delta$ most likely reflect an increase in growth rate.

In the core region, Ge contents are high and Al contents are low, as expected because Ge is compatible and Al incompatible. We find that Ge and Al concentrations in the core can both be modeled by $R\delta$ of $10^{-11}$ to $10^{-12}$ $m^2$ $s^{-1}$. In contrast, in the outer part of the crystal, both Ge and Al increase. If steady-state solutions are assumed, increasing $R\delta$ is needed to match the Al increase, while decreases in $R\delta$ are needed to match the Ge data.

In summary, our observations suggest that the crystal was initially growing at steady-state conditions (core white region),

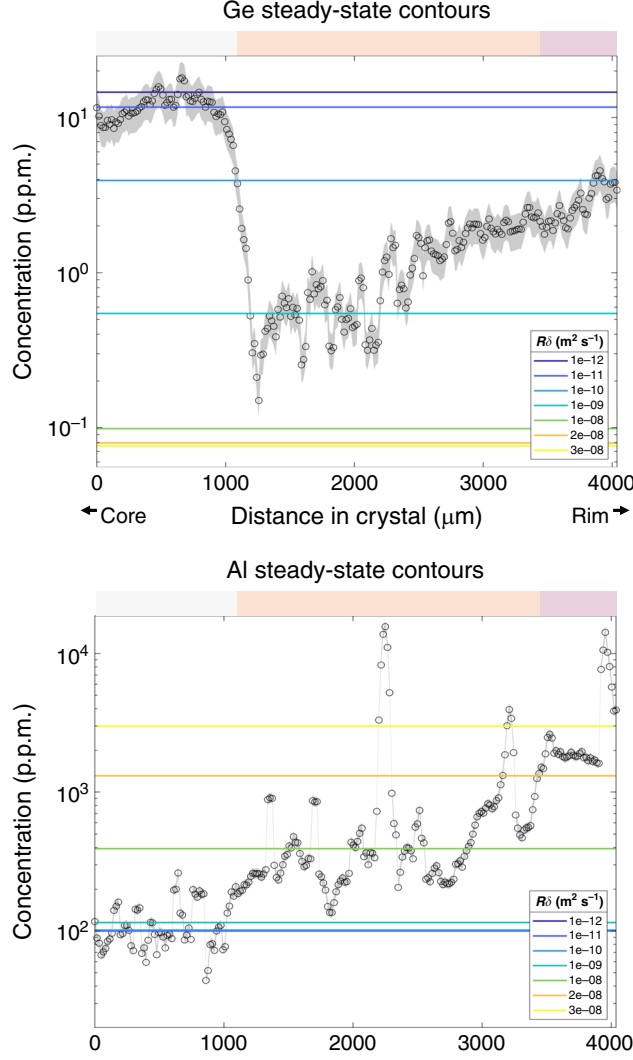

**Fig. 6 Comparison of measured profiles with modeled steady-state concentrations.** Steady-state concentrations calculated using Eq. 4 for different $R\delta$ and a constant $D$ of $8 \times 10^{-9}$ $m^2$ $s^{-1}$ for both Ge and Al. Data from both Ge and Al towards the core of the crystal correspond to similar $R\delta$, suggesting steady-state growth in the core. Beyond the core region, Ge and Al give different $R\delta$, suggesting a transient state. The lower $R\delta$ values ($10^{-10}$ to $10^{-12}$ $m^2$ $s^{-1}$) in the Al plot are nearly identical, causing them to plot on top of one another. Horizontal color bar corresponds to CL color regions. Error envelopes represent standard error of the mean based on $3\sigma$ of point analyses and a three-point moving average. The error values are 26% and 8.5% for Ge and Al, respectively.

but then experienced a sudden change in growth rate (at the white to orange transition). Thus, while the core elemental concentrations reflect steady crystal growth, subsequent elemental profiles reflect transient responses to sudden increases in the crystal growth.

As noted above, Ge and Al profiles in the outer part of the crystal are inconsistent with steady-state growth, and therefore, transient conditions must be considered. Using a given element, the ratio of initial to final growth rate can be constrained but not absolute growth rates. However, for two or more elements, growth rates can be constrained with Eq. 5 (with laser smearing accounted for in Supplementary note 1, Supplementary Eqs. 1 and 2, and Supplementary Fig. 9), which describes the response to an instantaneous increase in growth rate from a crystal initially

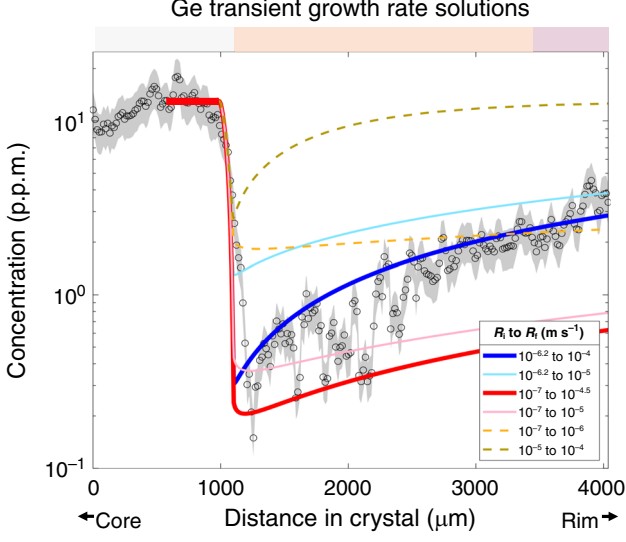

Ge transient growth rate solutions

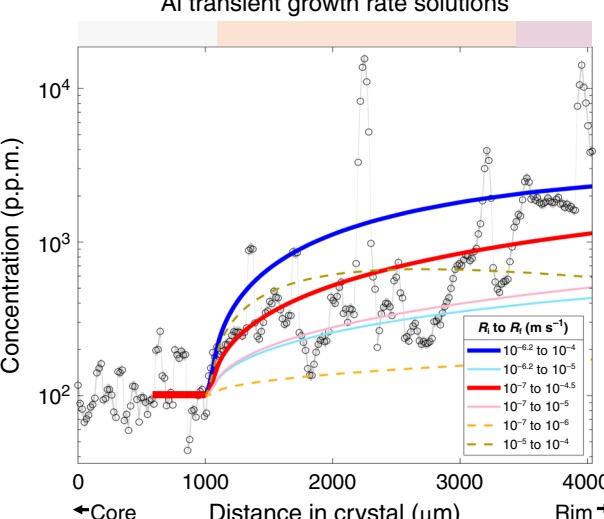

Al transient growth rate solutions

**Fig. 7 Comparison of measured profiles with modeled transient responses.** Solutions for transient responses for Ge and Al. Curves represent the response for a step function change in the crystal growth rate from $R_i$ to $R_f$ at ~1000 μm (the orange-white transition). A 2–2.5 order of magnitude change in growth rate is needed to match the data. Al should eventually reach a peak and fall back to steady state, yet the crystal did not grow long enough for this to occur. Color bar on top corresponds with CL color zones in crystal. Error envelopes represent standard error of the mean based on 3σ of point analyses and a three-point moving average. The error values are 26% and 8.5% for Ge and Al, respectively.

growing at steady state. We apply this equation to the transition from the white core zone to the orange middle zone because the constant profiles seen in the white core region suggest that steady state was attained as discussed above.

The two parameters we vary are the initial steady-state growth rate $R_i$ and the final growth rate $R_f$. We adopt the same diffusivities discussed above. $R_i$ and our assumed diffusivity control the steady-state concentration in the white core zone, while the ratio $R_f/R_i$ controls the shape of the response after an increase in growth rate. By varying initial growth rate between $10^{-7}$ and $10^{-6}$ m s$^{-1}$ and final growth between $10^{-5}$ and $10^{-4}$ m s$^{-1}$, the behaviors of both Al and Ge can be captured (Fig. 7). Adopting slower growth rates ($R_i = 10^{-7}$ m s$^{-1}$, $R_f = 10^{-6}$ m s$^{-1}$) results in too slow of a response and thus will not fit the curvature of the

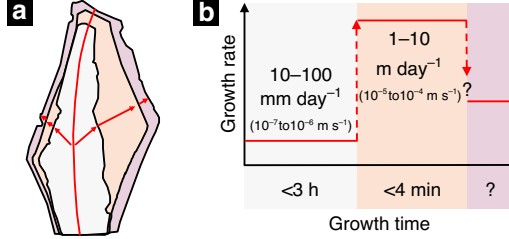

**Fig. 8 Cartoon depicting how crystal growth rates may have changed from core to rim. a** A sketch of the CL zonation of the crystal. **b** Reconstructed growth history of the crystal with overlain CL colors.

data within the confines of the model and its assumptions. Adopting faster rates ($R_i = 10^{-5}$ m s$^{-1}$, $R_f = 10^{-4}$ m s$^{-1}$) also does not fit the data well, showing too short response times. Uncertainties in diffusivities can be accounted for by adopting higher or lower (by an order of magnitude) diffusivities, resulting in higher and lower growth rates, respectively. If the partition coefficient for Al is lower (more incompatible) than used here, we predict nearly the same trace element profile. A more compatible partition coefficient (i.e., 0.1) will produce a slightly lower concentration profile, but still within the array of the data.

Based on the above analysis, we estimate that growth rate increased by ~100 times from the core to the rim, with initial growth rates on the order of 10–100 mm day$^{-1}$ ($10^{-7}$–$10^{-6}$ m s$^{-1}$) in the white core region to growth rates approaching 1–10 m day$^{-1}$ ($10^{-5}$–$10^{-4}$ m s$^{-1}$) in the orange middle region (Fig. 8). We are unable to estimate the growth rate of the outer purple zone because the orange middle zone never reached steady state, invalidating the application of Eq. 5 to the orange-purple transition. However, the slight decrease in Al (incompatible) and increase in Ge (compatible) in the purple outer zone suggests a slight decrease in growth rate.

Based on these estimates of crystal growth rate, we find that the white core region grew in <3 h, while the orange middle zone grew in <4 min. Ignoring the purple region, which is in any case thin, the entire crystal formed within hours. If these rapid crystal growth rates can be sustained, large decimeter-sized pegmatite crystals could have formed on the timescale of days. Similar rapid crystal growth rates have been inferred using independent approaches for orbicular granitoids[44].

For context, these crystal growth rates are compared to other geological rates in Fig. 9. Growth rates of pegmatitic quartz in this study are clearly fast compared to growth rates inferred for metamorphic garnets[45,46] and quartz in granitic plutons[47]. Quartz phenocrysts in volcanic rocks are thought to have also grown fast[48,49], but pegmatitic quartz growth rates are still at least as fast as the fastest inferred growth rates for volcanic quartz. Finally, crystal growth rates are compared with other geologic rates, such as plate motions[50] and earthquake-related deformation, such as slow slip and aseismic creep[51] (and references therein). This comparison allows us to contemplate whether crystal growth could play a significant role in other systems. We note that crystal growth timescales in pegmatitic systems can approach that of slow slip and aseismic creep, begging the question of whether crystal growth may be important during faulting or fault healing.

In advective systems, the diffusive boundary layer thickness is controlled by a balance between chemical diffusion in the fluid, which broadens the boundary layer, and advection, which thins the boundary layer[33,52,53]. The stronger the advective component, the narrower the boundary layer, thereby increasing the chemical gradient at the fluid–crystal contact and thus the flux of elements from the fluid into the crystal. Although Eq. 4 was derived for a

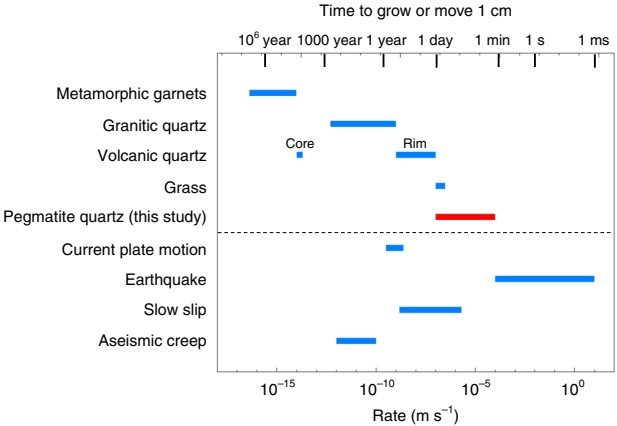

**Fig. 9 Comparison of pegmatite crystal growth rates from this study to rates of other geologic processes.** Data sources: metamorphic garnets[45,46]; granitic quartz[47]; volcanic quartz[48,49]; grass[62]; plate motions[50]; seismic rates[51] (and references therein).

static system, it can still be applied to a dynamic system at steady state. This is because chemical transport normal to the growing interface *within* the boundary layer is controlled by diffusion, not advection. This is analogous to thermal boundary layers in thermally convecting systems, where the thickness of the thermal boundary layer at steady state is controlled by convective thinning, but the vertical transport of heat within the boundary layer is controlled by thermal diffusion[54]. Thus, using the diffusivities adopted above and assuming the core region was near steady state (we cannot apply Eq. 4 to the orange or purple zones because they did not reach steady state), we find for the core region a boundary layer thickness of 10–100 μm. This assumes a steady state $R\delta$ ~$10^{-11}$–$10^{-12}$ m$^2$ s$^{-1}$ and $R$ ~$10^{-7}$ m s$^{-1}$ for the core, the latter constrained by modeling the transient response from the white core to the orange middle zones. We note that the formulation we used (Eq. 5) to model the transient response was derived for a static case. In dynamic systems, boundary layers would be thinned, and because growth rate scales inversely with boundary layer thickness in Eq. 5, our estimated growth rates from the static transient model are minimum bounds. Thus, the boundary layer thickness in the core is a maximum bound, and importantly, is considerably smaller than that for the static case ($\delta \sim D/R$ ~8 cm). Advective thinning of the boundary layer is thus required.

In an advecting system, $\delta$ is related to fluid velocity[55]. The chemical boundary layer thickness $\delta$ scales inversely with the Sherwood number Sh, that is, $\delta \sim L/$Sh, where $L$ is the characteristic crystal length and Sh relates convective mass transfer to diffusive mass transfer[56]. Sh is related to the Reynolds number Re of the fluid (see Supplementary note 2 along with Supplementary Eqs. 3–5[55] and Supplementary Table 1), which relates inertial to viscous forces. Re is given by $UL\rho/\mu$, where $U$ is the fluid velocity, $\rho$ is the fluid density, and $\mu$ is the fluid viscosity[57]. Our estimated boundary layer thickness corresponds with a Re of 20–4000 and an average fluid velocity of 0.1–20 cm s$^{-1}$. Since changes in diffusivity cause a proportionate change in growth rate, the change will cancel out, meaning this value holds for all Ge diffusivity values. Given that Re > ~2300 in pipe flow is considered turbulent[57], our quartz crystals may have grown in a highly dynamic environment. For comparable conditions, but in silicate melt systems, Re for crystal growth is «1 due to the very high viscosities of silicate melts.

Pegmatites appear to form on much shorter timescales than typical plutonic rocks, but paradoxically, they have much larger crystals, which alone requires that crystal growth rates must be much higher in pegmatitic systems compared to typical plutonic

systems. Exactly why is unclear, but it is widely thought that an abundance of a free fluid phase is important for the formation of pegmatites[8,38,39]. The reasoning is that diffusivities in fluids are higher than that in silicate melts, thereby permitting more rapid flux of nutrients to the crystal and enhancing growth rates[58,59]. However, an increase in diffusivity alone may not be enough to decrease growth times from thousands of years to hours as diffusivities of common cations in water at elevated temperature are on the order of $10^{-8}$ m$^2$ s$^{-1}$[42,43] and diffusivities in silicate melts are on the order of $10^{-10}$ m$^2$ s$^{-1}$. As discussed above, flux to the crystal can also be increased by decreasing the thickness of the chemical boundary layer in the fluid via advection. Our work shows that the boundary layer thickness of these crystals was ~$10^4$ times thinner than a purely diffusive boundary layer with no advective thinning, ultimately resulting in a $10^4$-fold enhancement of growth rates. Such thin boundary layers, as concluded above, are maintained only under turbulent conditions.

These findings may shed light on the nature of mineralization in the Stewart pegmatite. We envision the pegmatite originating by dike emplacement of a hydrous silicate magma into the host gabbro rock. The outer zones (upper and lower) of the pegmatite body solidified first by cooling against already solidified host gabbro, while the central core zone solidified last as the crystallization front progressed inwards. This progressive crystallization concentrates volatiles and other incompatible elements/components in the residual silicate melt in the core zone. Eventual saturation in these volatiles leads to exsolution of a free fluid phase and a buildup of overpressure, triggering upwards fracturing and transport of the fluid phase into the solid upper zone and generating chimneys and miarolitic cavities.

As the fluid expands into the cavities, the resulting decompression leads to supersaturation of dissolved constituents in the fluid, driving rapid nucleation and crystal growth. Changes in pressure may also cause rapid changes in growth rates. Emplacement of these fluids into fractures appears be accompanied by turbulence, which significantly enhances crystal growth rates, allowing for pegmatitic crystals to form within hours. We thus envision miarolitic cavities to have formed under highly dynamic and ephemeral conditions.

Our crystal growth rates clearly only reflect processes operating in the formation of late-stage miarolitic cavities and do not apply to crystallization in other parts of the pegmatite. Similar studies using trace element chemistry of crystals in different parts of a pegmatite body should be measured in the future to better understand how crystal growth rates vary over the entire history of a pegmatite. In summary, our results suggest that it may be worth reconsidering whether large crystals in other magmatic systems, such as porphyries, are a consequence of rapid growth rather than by slow growth over long incubation times.

## Methods

The transient response to a sudden change in the crystal growth after growing at steady state is described by Smith et al.[34]:

$$C_s = C_{eq} \left( \begin{array}{c} 1 - \frac{1}{2}\text{erfc}\left(\sqrt{\frac{R_f x'}{4D}}\right) + (1-k)\left(\frac{\frac{1}{2}-\frac{R_i}{R_f}}{k-\frac{R_i}{R_f}}\right)\exp\left(-\frac{R_i}{R_f}\left(1-\frac{R_i}{R_f}\right)\frac{R_f}{D}x'\right)\text{erfc} \\ \left(\left(\frac{R_i}{R_f}-\frac{1}{2}\right)\sqrt{\frac{R_f x'}{D}}\right) + \frac{2k-1}{2}\left(\frac{1-\frac{R_i}{R_f}}{k-\frac{R_i}{R_f}}\right)\exp\left(-k(1-k)\frac{R_f}{D}x'\right)\text{erfc}\left(\left(k-\frac{1}{2}\right)\sqrt{\frac{R_f x'}{D}}\right) \end{array} \right),$$

(5)

wherein time has been transformed to distance $x'$ within the crystal as that is the quantity that can be measured ($x' = R_f \times t$). Meanings of other symbols are outlined earlier in the text. This equation has three terms on the right-hand side (we consider $1 - \frac{1}{2}\text{erfc}\left(\sqrt{\frac{R_f x'}{4D}}\right)$ to be one term). The first term is most important at long timescales as the other two terms decay exponentially with time. This first term describes the long-term diffusional response and only depends on the final growth rate $R_f$ because on long timescales, the initial condition does not matter. The second and third terms are important on short timescales and are thus sensitive to the initial condition, which is controlled by $R_i$, $D$, and $k$. The second and

third terms depend on $R_i$ and $R_f$, but the first term only depends on $R_f$ which allows determination of crystal growth rates provided we have constraints from two elements, such as Ge and Al.

## Data availability

All data generated in this study can be found in this published article or in Supplementary Data 1.

## Code availability

All models were calculated using off the shelf software (MATLAB) and using the equations presented in this article and its Supplementary information.

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

## Acknowledgements

This work was supported by NSF-EAR 1753599 to C.-T.A.L. Discussions with Hehe Jiang, Ming Tang, Eytan Sharton-Bierig, Julin Zhang, and Blue Sheppard are appreciated.

## Author contributions

P.R.P. analyzed samples, provided interpretation, and was involved in writing. C.-T.A.L. helped with interpretation and writing. D.M.M. supplied samples and geologic context.

## Competing interests

The authors declare no competing interests.
