## [Peer Review File · Nature Communications]

Reviewers' comments:

Reviewer #1 (Remarks to the Author):

Manuscript NCOMMS-20-02845 by Phelps et al. address the intriguing problem of mineral growth rates in LCT pegmatites. It is a classic discussion The subject has strong implication on several open questions in geosciences (composition of pegmatite growing media, distribution and concentration of strategic metals in pegmatite bodies, element diffusion and partitioning, etc.) and thus it represents a relevant topic to be published on this journal.

The manuscript is well written (anyhow consider that I'm not mother tongue) and organised as well as analytical data and kinetic calculation seem to be robust.

However, I am very perplexed about the position of the pockets (chimneys) and the texture of the quartz crystals chosen for the study.

The authors claim to have sampled the quartz crystals in a cavity that cuts the upper perthitic zone of the pegmatite, well outside the core zone where the largest and most beautiful quartz, elbaite and k-feldspar crystals are usually found in pegmatites (Lines 70-75; Figure 1). Such chimneys looks like a relatively late feature of this pegmatite body since they cut the primary mineralogical banding of the pegmatite body. Similar structures are found, although different in geometry/attitude, in many LCT pegmatites worldwide and usually show clear evidences of hydraulic fracturing and hydrothermal replacement of earlier mineral phases. Although miarolitic pockets in chimneys contain euhedral crystals of elbaite (typical of core-zone pockets), their attitude and position in the pegmatite body seem to be related to local and peculiar structural conditions experienced by the Stewart LCT pegmatite. Thus, the study is not based on those crystals which, on average, characterise the primary stage of formation of LCT pegmatite bodies.

The above criticism is reinforced by the texture of the quartz crystals analyzed. As highlighted by the authors (Lines 85-86), "they are peculiar in that they have flattened crystal habits with the {1000} axis being short" (although curly brackets are commonly used for crystallographic forms, not for axes).

Quartz crystals in pegmatite pockets are usually prismatic, more or less elongated on c axis. The typical habit of alpha quartz. Unusual flat, frequently irregular, quartz crystals are commonly found in pegmatite pockets that experienced hydraulic fracturing. In such cases, flat shards of pristine pegmatite quartz may recrystallize being overgrown by a late quartz layer responsible for the final "euhedral", yet irregular and odd, shape of the crystals.

Due to the lack of a detailed description of the pocket structure and crystal morphology, I cannot be sure my criticism is correct 100%. However, CL images (es. Figure 2) show an irregular white core (in fact it resembles a typical concave-shaped splinter) overgrown by late layers that, progressively, tend to take a more regular, euhedral shape. The large increase of calculated growth rates from such core to the external rims maybe is true but should be attributed to very distinct stages of crystallization, from truly early pegmatitic to late hydrothermal.

Again, results are probably correct but from the title the reader would expect to find data and calculation performed on common pegmatitic crystals not odd crystals that experienced multiple and complex crystallization events.

In spite of the excellent analytical work, the geo-petrological interpretation suffers from a choice of samples that is not adequate for solving the problem reported in the introduction and in the title of the manuscript. It is my opinion that this study would be great if similar data are provided also on "normal" quartz crystals extracted from core-zone pockets. In order to generalize the interpretation, as it would seem to understand from the title, other LCT pegmatites from different areas/ages should be investigated.

Finally, pockets volume and the mass of crystals forming inside the pockets represent a small fraction of a pegmatite body. Quartz crystallizes from the very beginning of the formation of a pegmatite intrusion and growth rates should be calculated through the whole pegmatite history before to generalize the intriguing conclusion of this contribution.

It is up to the publisher to decide whether the history of growth-rate changes presented in this manuscript is adequate for publication in this journal.

Andrea Dini

Reviewer #2 (Remarks to the Author):

This paper concerns an interesting conundrum: why do pegmatites have such exceedingly coarse grain sizes when they are thought to have cooled extremely rapidly? In this contribution, the authors look at zoning in pegmatitic quartz and suggest that extremely rapid crystal growth occurs in the turbulent, fluid-rich environments. This paper is timely, as the effects of kinetics (i.e., crystal growth rates and diffusion) on crystal compositions is increasingly becoming a problem that people are considering in studies of the magmatic processes and timescales revealed by crystal compositions.

Overall, I think this is a nice paper, and the authors do a nice job of laying out the boundary layer problem. I do, however, think it would be better suited for a short-format topical journal, such as *Geology*.

I have a few general comments (below) and some suggested comments and edits in the attached PDF:

1. The focus of this paper is on coarse-grained pegmatites, thought to have crystallized from evolved magmas or fluids fractionated late in the history of pluton building. Aplites often have similar compositions to pegmatites and are also thought to crystallize from similar late-stage fractionated liquids, yet they have very different textures – they are very fine-grained. In some cases, aplites and pegmatites are even associated (I'm not sure if this is the case in the Stewart pegmatite). I would like to see the authors speak to this discrepancy at some point in the paper.
2. In Figure 2a, I see a boundary within the white core – a thin, brighter white rim around the core, between the core and the orange zone. Can the authors give any idea of what this represents? It doesn't pop out in the geochemical data for transect A-A', but was this boundary sampled? I see two lines of spots in the crystal, neither of which corresponds with A-A' – were these laser spots? The density of spots does not correspond with the amount of data that appears to comprise A-A', so I am not sure what these are, but if they are, they don't appear to be sampling the dark-bright white boundary.
3. Were transects taken parallel {0001} at all? Can you say anything about anisotropy in growth rates?
4. I would like to see growth rates given in m/s, along with the mm/day or m/day values. This makes it easier to compare with other rates. For example, 10-100 mm/day is equivalent to 10^{-6} - 10^{-7} m/s, which are equivalent to rates suggested for rim growth of volcanic quartz (Gualda & Sutton 2016). I realize this is depicted in Figure 9, but I would also like to see it in the text and abstract.

Reviewer #3 (Remarks to the Author):

Summary

This paper is provocative because it claims that crystals in pegmatites can grow at rates of 1-10 m/day. This is 3 orders of magnitude faster than KDP crystals grown in the labs at Livermore for the NIF project where they engineered the hydrodynamic conditions to promote rapid growth (Zaitseva et al., 1999). The data upon which this conclusion is reached are trace element profiles (4 mm long) within a cm-sized quartz crystal that show a sharp, approximately two orders of magnitude drop in [Ge] that is correlated with a sharp, approximately one order of magnitude increase in [Li]. Following the sharp drop, there is a steady increase in [Ge], [Li], and [Al] throughout the "orangish middle zone." The sharp drop is interpreted to be due to a sudden change in growth rate, and the gradual increase is inferred to be a time-dependent recovery to a new steady state. These interpretations guide the development of a quantitative model. The authors work through it and show that if the assumptions that underlie the model are to be believed, it leads to the conclusion that the gray zone grew at 10-100 mm/day and the orange zone grew at an astounding rate of 1-10 m/day and could thus have formed within minutes. This paper is suitable for a Nature journal if the authors can further convince the reader that the 1-10 m/day number is inescapable; i.e., that it can't be attributed in part to other factors or considerations. As it stands, I'm not yet convinced for the following reasons:

- (1) In the model, the initial and final growth rates are specified and not tied to any growth law related to supersaturation. But, a sharp drop in Ge between "grey zone" and "orange zone" implies, by the author's reasoning of similar partitioning behavior of Ge and Si, a sharp drop in Si. If conditions are isothermal and isobaric, shouldn't this mean a decrease in supersaturation, and by extension, decrease in growth rate?
- (2) The inferred temperature is taken from the Wark and Watson (2006) TitaniQ thermometer, which is only applicable at 1000 bars. The follow-up calibration by Thomas et al. (2010) shows that there is a strong pressure-dependence, and since the inferred pressures are much lower (150-200 bars), the inferred temperature of 600C needs to be updated. This will affect the diffusivities of the modeled elements by different amounts because they each have a different activation energy.
- (3) Following on (2), there is debate as to whether it is safe to extrapolate the Thomas et al. (2010) expressions to pressures lower than 500 bars because Huang and Audetat (2012) got very different results at low pressure. Results from both calibrations should be compared and the implications considered thoughtfully.
- (4) Regardless of which calibration is used, please address the appropriateness of using a trace element thermobarometer, which is based on principles of equilibrium trace element partitioning, while arguing at the same time that trace element uptake is kinetic.
- (5) The possibility of pressure fluctuations in the crystallizing environment is not considered, but could have a large impact on trace element uptake and also explain the sharp changes in growth rate and trace element concentration, or even trace element concentration at constant growth rate. This may actually work to the author's advantage if they can argue that pressure doesn't affect K_d very much but can lead to sharp change in growth rate.
- (6) Rates should be directly compared to those obtained by Webber et al. (1999). They estimated 10-5 cm/s (0.86 cm/day) for the larger 10 cm long tourmaline crystals. The largest quartz crystal analyzed (in the supp) has a long dimension of about 10 cm according to line 84. Did these quartz crystals grow 2-3 orders of magnitude faster than the adjacent tourmalines? If so, why the difference?
- (7) The partition coefficients are treated as constant even though they are known to strongly depend on growth rate (e.g., Huang and Audetat, 2012 for Ti in quartz) as well as the presence of other impurities (e.g., Efremova et al., 2004 for multiple elements in KDP crystals). This could have a significant impact on the inferred growth rates and conclusions. How much wiggle room is there? If one of the K_d 's were off by an order of magnitude, or if the K_d 's were coupled, how would that impact the results?

(8) In the limit that the boundary layer is infinitely thin, the growth is still rate-limited by nutrient attachment kinetics. In an aqueous solution, for example, I imagine that the Si^{4+} ions (or compounds?) would have a hydration sphere that needs to be shed prior to incorporation into the crystal. I encourage the authors to look into this and check to make sure that the linear growth rates of 1-10 m/day can be accommodated by ion desolvation rates.

(9) The extreme growth rates imply high supersaturation. If so, what prevents homogeneous or heterogeneous nucleation of many crystals? (which would bring the supersaturation down)

All that aside, I like the paper and the development of new ideas therein. The conclusions, if they stand up, would certainly be of broad interest and would make for neat headlines. I suspect that the authors may be able to address most, if not all, of these comments above and below and hope that they get the opportunity to do so.

References

Efremova, E., Sukhanovskaya, A., and Kuznetsov, V., 2004, Effective Distribution Coefficients of Cation Impurities, in KDP Crystals, Inorganic Materials, v. 40, p. 636-640.

Zaitseva, N., Carman, L., Smolsky, I., Torres, R., and Yan, M., 1999, The effect of impurities and supersaturation on the rapid growth of KDP crystals, Journal of Crystal Growth, p. 512-524.

Minor comments:

Line 25: Kinetic theory was \diamond The equations of crystal growth and trace element uptake are

Line 27: delete anomalously

Line 30: shown \diamond inferred

Line 51: anomalously rapid crystal growth \diamond rapid crystal growth at high supersaturation

Line 53: "Does rapid crystal growth influence the dynamics of magmatic systems?" is not addressed later and should be deleted or addressed.

Line 66: I'm confused here. Did the pegmatite form from a dike that intruded the gabbro or from fluids exsolved and fractionated from the gabbro?

Line 68: 50 m wide dike – Webber et al. refer to this one as a 25 m wide dike. Why the difference?

Line 71: miarolitic is a bit jargon-y and could be defined using a parenthetical

Line 78: A couple more details on the conductive cooling model by Weber would be helpful.

Starting T? Latent heat? Instantaneous emplacement following by cooling? No heat from below?

Line 80: This is only an upper bound on cooling to 550C. If crystallization can occur to lower temperatures, which is possible if not likely because crystals are growing in a low viscosity Re fluid phase, then the timescales could be much longer.

Line 82: We investigated [three] quartz crystals

Line 83: delete 'perfectly'

Line 84: range in size from a few cm to 10 cm – since there are only 3 you can give the long dimension of each.

Line 108: "forcing the sum of all measured metals to equal 100%." Should I worry about this introducing artifacts?

Line 109: One crystal out of three is not "representative." Or, if it is representative, please state what it is representative of; e.g., the width and amplitude of trace element variations that you see in all three crystals?

Line 113: why use the word likely here when the information is available to state whether or not the CL variations correlate with trace element concentration?

Line 116: Here you state that transitions in CL correlate with trace element concentrations. So, maybe delete line 113?

Line 113: Why not pressure cycling?

Line 115: Are these really representative? If so, how so? At a glance the transects in the supplement look quite different from the one chosen to be the focus of the main paper.

Line 136: T = 600 C may need to be revised (see comments above about TitaniQ calibrations)

Line 137: where does the TiO_2 activity = 0.6 come from???

Line 140: "...would require nearly 2 My to generate the Al profiles by diffusion [assuming an initial step function]" what aspect of the profiles? (What length scale of heterogeneity?)

Line 142: I would move the first sentence to the end of the previous paragraph. The topic sentence here should be about Li diffusion.

Line 142: "presumably" ?

Line 143: diffusivities \diamond diffusivity

Line 143: predict \diamond generate or yield

Line 151: observed \diamond inferred

Line 155: another possibility is changes in pressure, which can be rapid and variable

Line 158: that \diamond those

Line 159: that \diamond those

Line 160: please provide a Kd value for Al in quartz here

Line 161: "high Al contents [assuming equilibrium partitioning]"

Line 177: As far as I can tell, the Watson and Muller (ref. 31) paper is only cited once: "In this section, we explore how changes in crystal growth rate can affect how trace elements are incorporated into crystals^{29–31}." It'd be a good idea to give that paper a bit more credit for the ideas between lines 177-246 because Watson and Muller also discuss crystal growth in the same framework and with consideration for how an advecting fluid could decrease the BL thickness and so forth.

Line 179: Usually partition coefficients are denoted by capital K and lowercase k is reserved for rate constants. I don't really care about this, so as long as it's consistent throughout, the authors can do as they please.

Line 179: also, I suggest changing "equilibrium partition coefficient" to "partitioning behavior"

Line 228: "For very large Peclet numbers, such as for large growth rates, the crystal composition C_s approaches the far-field liquid composition C_{eq} ." This was confusing to me at first – might be helpful to point out that C_{eq} is not the equilibrium concentration in the crystal, or call C_{eq} something else.

Lines 231-236: Here would be a good place to re-cite Watson and Muller (ref 31), for example.

Line 243: delete "lower"

Lines 248-290: I understand why the authors went the route of an analysis of different steady states, but the text doesn't fill in the blanks sufficiently for the uninitiated. Maybe add a sentence to introduce the value of examining the steady state cases.

Line 260: The inferred $T = 600C$ is stated again, highlighting the importance of this assumption.

Line 272: "must" is too strong of a word here because a certain degree of oversaturation (disequilibrium) is required for nucleation and growth.

Line 294: What is meant by "models account for laser smearing"?

Line 296: For \diamond Using or "A given element can be used to constrain ..."

Lines 296-299: This may be the most important statement in the paper, and yet, it feels buried.

Line 303: here would be an appropriate place to add the caveat "curvature of the data, within the confines of the model and its assumptions."

Line 304: Uncertainties in D's are not accounted for, per se. I would say something like " higher or lower growth rates could be accommodated to some extent by uncertainties in diffusivities."

Line 307: too strong wording. Suggest changing "at least" to "some"

[around 309]: Here is the

Fig 2: No need to show B-B' in the figure since it is not discussed in the main text. Please annotate within the panels where the spikes in Li and Al are due to inclusions.

Fig. 7: This is actually the neatest figure herein, but there's still a signal superimposed on the transient recovery after the perturbation. Thoughts on this?

Supp Fig. 1. What are the vertical dashed lines for?

We thank the reviewers for their critical feedback. The manuscript has been significantly improved because of it.

Response to Reviewer #1's comments:

We appreciate Andrea Dini's critical comments related to the nature of the chimneys and the quartz. We have made changes to the manuscript we believe are necessary to address his comments.

- “However, I am very perplexed about the position of the pockets (chimneys) and the texture of the quartz crystals chosen for the study.

“The authors claim to have sampled the quartz crystals in a cavity that cuts the upper perthitic zone of the pegmatite, well outside the core zone where the largest and most beautiful quartz, elbaite and k-feldspar crystals are usually found in pegmatites (Lines 70-75; Figure 1). Such chimneys looks like a relatively late feature of this pegmatite body since they cut the primary mineralogical banding of the pegmatite body. Similar structures are found, although different in geometry/attitude, in many LCT pegmatites worldwide and usually show clear evidences of hydraulic fracturing and hydrothermal replacement of earlier mineral phases. Although miarolitic pockets in chimneys contain euhedral crystals of elbaite (typical of core-zone pockets), their attitude and position in the pegmatite body seem to be related to local and peculiar structural conditions experienced by the Stewart LCT pegmatite. Thus, the study is not based on those crystals which, on average, characterise the primary stage of formation of LCT pegmatite bodies.”

We agree with some of these concerns and have noted this in the paper now. We would like to note that the chimneys make up ~50% by volume of the perthite zone and many of the large tourmalines, in fact, are found in the chimneys (see Morton et al., 2018). The chimneys may be “late”, but only in that they do not form during the emplacement of the pegmatite and instead after the outer parts have already solidified. Importantly, the chimneys are forming when the core zone is still active. The chimneys are clearly a key step of the entire evolution of this pegmatite body. We do not necessarily expect our calculated rates of crystallization to apply to crystal growth throughout the entire pegmatite body. But what they do show is that rapid crystal growth rates can happen.

- “The above criticism is reinforced by the texture of the quartz crystals analyzed. As highlighted by the authors (Lines 85-86), “they are peculiar in that they have flattened crystal habits with the {1000} axis being short” (although curly brackets are commonly used for crystallographic forms, not for axes). Quartz crystals in pegmatite pockets are usually prismatic, more or less elongated on c axis. The typical habit of alpha quartz. Unusual flat, frequently irregular, quartz crystals are commonly found in pegmatite pockets that experienced hydraulic fracturing. In such cases, flat shards of pristine pegmatite quartz may recrystallize being overgrown by a late quartz layer responsible for the final “euhedral”, yet irregular and odd, shape of the crystals.

“Due to the lack of a detailed description of the pocket structure and crystal morphology, I cannot be sure my criticism is correct 100%. However, CL images (es. Figure 2) show an irregular white core (in fact it resembles a typical concave-shaped splinter) overgrown by late layers that, progressively, tend to take a more regular, euhedral shape. The large increase of calculated growth rates from such core to the external rims maybe is true but should be attributed to very distinct stages of crystallization, from truly early pegmatitic to late hydrothermal.”

This is a valid point that we had not thought about before. We thank the reviewer for raising this issue. Our various CL images do not seem to support the cores being shards, but even if they are shards, our results still hold. Our models start in the slightly brighter white growth layer that surrounds the core. In other words, our calculations of growth are for the middle layer, not the core. Our calculations only assume there has been new growth around the core. Whether it is a shard or not is not critical because growth of the new layer initiates instantly. That being said, since the chemistry does not change going from the slightly dimmer core towards the brighter, white zone, this leads us to believe the growth rates were very similar. Also, thank you for noticing our error related to the axis notation. We have rectified it in the text.

- “Again, results are probably correct but from the title the reader would expect to find data and calculation performed on common pegmatitic crystals not odd crystals that experienced multiple and complex crystallization events. In spite of the excellent analytical work, the geo-petrological interpretation suffers from a choice of samples that is not adequate for solving the problem reported in the introduction and in the title of the manuscript. It is my opinion that this study would be great if similar data are provided also on “normal” quartz crystals extracted from core-zone pockets. In order to generalize the interpretation, as it would seem to understand from the title, other LCT pegmatites from different areas/ages should be investigated.”

This is a valid critique, but we do not think we over-generalized in our title. We believe any attempt to calculate growth rates within pegmatites is a positive contribution to our understanding of these fascinating geological formations. We hope we are not misleading readers into believing these rates describe the entire history and have included text in the manuscript now to discourage this interpretation. We agree and think it is a great idea to look at more quartz and other pegmatites, but we believe this is beyond the scope of this paper. Our goal (and title) were (are) not meant to imply that these crystal growths describe the lifetime of a pegmatite or even its evolution. Rather, we only want to show that crystallization can occur rapidly within pegmatites. This does not by itself imply that pegmatites themselves crystallize quickly as the evolution of a pegmatite may consist of multiple stages of crystallization.

- Finally, pockets volume and the mass of crystals forming inside the pockets represent a small fraction of a pegmatite body. Quartz crystallizes from the very beginning of the formation of a pegmatite intrusion and growth rates should be calculated through the whole pegmatite history before to generalize the intriguing conclusion of this contribution.

As mentioned in our first comment, the chimneys make up ~50% by volume of the perthite zone, which is not an insignificant portion. Also, many of the gem quality elbaite that characterize these types of pegmatites are found in these chimneys. We believe this is an important stage of the formation history to characterize. We completely agree with the reviewer that we should track quartz crystallization throughout all parts of the pegmatite and through its entire history. This is however beyond the scope of this paper. Our paper only shows that rapid growth can happen, and thus our paper should be used as motivation to pursue exactly what the author suggests, but this is better for the entire community to pursue, including ourselves.

Response to Reviewer #2's comments:

We appreciate this reviewer's critiques and questions, and we hope we have done a suitable job addressing them.

- “Overall, I think this is a nice paper, and the authors do a nice job of laying out the boundary layer problem. I do, however, think it would be better suited for a short-format topical journal, such as *Geology*.”

We thank the reviewer for their kind words. We also agree this could have gone into a short-format journal. However, we wanted to be sure we had enough space to lay out the proper math for the problem and have a complete explanation. Plus, we have already submitted here to *Nature Communications*. While we would love to shorten it and submit to a short format journal like *Geology* or *Nature Geoscience*, we do not think we could do justice to the paper in such a short format.

- “The focus of this paper is on coarse-grained pegmatites, thought to have crystallized from evolved magmas or fluids fractionated late in the history of pluton building. Aplites often have similar compositions to pegmatites and are also thought to crystallize from similar late-stage fractionated liquids, yet they have very different textures – they are very fine-grained. In some cases, aplites and pegmatites are even associated (I'm not sure if this is the case in the Stewart pegmatite). I would like to see the authors speak to this discrepancy at some point in the paper.”

We agree that this is an intriguing question. However, we do not think the work we have done in this paper points towards an answer to this question. Ultimately, the difference between aplites and pegmatites is whether growth or nucleation dominates. In pegmatites, there is more growth, leading to larger crystals, while in aplite, there is more nucleation. A separate study would need to be done to parse out why growth seems to overwhelm nucleation in this pegmatite. That is not to say nucleation does not occur (otherwise there would be no crystals), but it does not dominate.

- “In Figure 2a, I see a boundary within the white core – a thin, brighter white rim around the core, between the core and the orange zone. Can the authors give any idea of what this represents? It doesn't pop out in the geochemical data for transect A-A', but was this boundary sampled? I see two lines of spots in the crystal, neither of which corresponds with A-A' – were these laser spots? The density of spots does not correspond with the amount of data that appears to comprise A-A', so I am not sure what these are, but if they are, they don't appear to be sampling the dark-bright white boundary.”

The reviewer has a good idea. There is a slightly brighter rim surrounding the slightly dimmer core regime. We did sample both regions with our transects, but not with our point analyses. The A-A' transect corresponds with where we dragged the laser across the crystal rather than using points. This is spoken about in the paper. We assume this brighter white rim to be part of the core area because there is no chemical change across the boundary. We have revised the text to make a point about the boundary. Since there is no chemical change, we suspect the growth rates are very similar between the two zones.

- “Were transects taken parallel {0001} at all? Can you say anything about anisotropy in growth rates?”

No, there were no transects taken parallel to the {0001}, only parallel to the {1000}. However, this sort of study could be applied to the {0001} to determine its growth rate, and therefore the growth rate anisotropy. This would be an interesting future study.

- “I would like to see growth rates given in m/s, along with the mm/day or m/day values. This makes it easier to compare with other rates. For example, 10-100 mm/day is equivalent to 10^{-6} - 10^{-7} m/s, which are equivalent to rates suggested for rim growth of volcanic quartz (Gualda & Sutton 2016). I realize this is depicted in Figure 9, but I would also like to see it in the text and abstract.”

Changed accordingly. Thank you for catching that.

Response to Review #3's comments:

We also appreciate this reviewer's comments. They have given us a lot to think about and have helped make our paper better.

- In the model, the initial and final growth rates are specified and not tied to any growth law related to supersaturation. But, a sharp drop in Ge between “grey zone” and “orange zone” implies, by the author's reasoning of similar partitioning behavior of Ge and Si, a sharp drop in Si. If conditions are isothermal and isobaric, shouldn't this mean a decrease in supersaturation, and by extension, decrease in growth rate?

The reviewer is right in pointing out that there will be a decrease in Si content as well as supersaturation accompanying a drop in Ge. Supersaturation, however, does not set the growth rate. It provides the driver for growth. What determines growth rate is a combination of reaction kinetics at the surface of the crystal, the supply rate (diffusion) of the mineral constituents (Si in the case of quartz), and the supersaturation. If we assume the Si content is still supersaturated at the surface, even though the degree of supersaturation dropped, then growth will continue, and diffusion will sort out the Si concentration gradients set up from the changes in growth rate. This is why we look at trace elements rather than major elements. We can circumvent unknowns related to the supersaturation or the reaction kinetics, especially considering how little we know about how the system is evolving. Also, if the growth rate was decreasing, we would expect to see a significantly different trace element profile. We have clarified accordingly in the text.

- “The inferred temperature is taken from the Wark and Watson (2006) TitaniQ thermometer, which is only applicable at 1000 bars. The follow-up calibration by Thomas et al. (2010) shows that there is a strong pressure-dependence, and since the inferred pressures are much lower (150-200 bars), the inferred temperature of 600°C needs to be updated. This will affect the diffusivities of the modeled elements by different amounts because they each have a different activation energy.”

There is a pressure dependence and certainly dropping pressures to 100 bars is significant. However, the pressures of concern here are in the kbar regime. In any case, using the thermobarometer from Huang & Audetat (2012) who did experiments at similar T and P to our system (600-800°C, 1-10 kbar), the temperature can be updated to 540°C at 200 MPa (2 kbar). This value is very similar to what we calculate using Wark & Watson's original thermometer for a TiO₂ activity of 1. Huang & Audetat do not include effects of Ti activity, though. Therefore, this temperature will get higher with lower activity. For the sake of argument, we can use this lower temperature and see how it changes our results. The diffusivity goes from $8 \times 10^{-9} \text{ m}^2/\text{s}$ to $7.4 \times 10^{-9} \text{ m}^2/\text{s}$ for Al. Considering we are looking for order of magnitude estimates for the growth rate, this value will not change our answer significantly. Using the calibration from Thomas et al. (2010) gives a lower temperature of 460°C for an activity of 0.6. This temperature drops the diffusivity to $6.4 \times 10^{-9} \text{ m}^2/\text{s}$, which is still a small difference and not a significant change.

- “Following on (2), there is debate as to whether it is safe to extrapolate the Thomas et al. (2010) expressions to pressures lower than 500 bars because Huang and Audetat (2012) got very different results at low pressure. Results from both calibrations should be

compared and the implications considered thoughtfully.”

This is not a problem in our system, since we are using pressures of 200 MPa, which equates to 2 kbar not 100 bars. See note above for the comparison.

- “Regardless of which calibration is used, please address the appropriateness of using a trace element thermobarometer, which is based on principles of equilibrium trace element partitioning, while arguing at the same time that trace element uptake is kinetic.”

This is a great point that we have considered. We realize this issue and even cite a study that brings up this issue (Pamukcu et al. 2016). We only used Ti values in the core of the crystal to estimate temperature. These values should be close to the equilibrium value because the low Ti values suggest slow growth. If nothing else, the core was growing more slowly, so the kinetic effects will not be as large. This is the best temperature estimate we could determine based on the data we could have collected and the current state of the literature.

- “The possibility of pressure fluctuations in the crystallizing environment is not considered, but could have a large impact on trace element uptake and also explain the sharp changes in growth rate and trace element concentration, or even trace element concentration at constant growth rate. This may actually work to the author’s advantage if they can argue that pressure doesn’t affect K_d very much but can lead to sharp change in growth rate.”

We agree completely with the reviewer on this point. We think the opening of the chimneys causes a pressure drop, driving changes in crystallization. Other changes may have occurred as well, but we do not have a distinct mechanism for what causes them (see lines 403-408).

- “Rates should be directly compared to those obtained by Webber et al. (1999). They estimated 10-5 cm/s (0.86 cm/day) for the larger 10 cm long tourmaline crystals. The largest quartz crystal analyzed (in the supp) has a long dimension of about 10 cm according to line 84. Did these quartz crystals grow 2-3 orders of magnitude faster than the adjacent tourmalines? If so, why the difference?”

Webber et al. (1999) reports cooling timescales for the Stewart pegmatite as well as the Himalaya dike among others in the area. There are no crystal growth rates reported for the Stewart pegmatite. The crystal growth rates reported in Webber et al. are for the Himalaya dike, a 1 m thick dike in the same county as the Stewart pegmatite but a different mining district. The Stewart is significantly larger at 50 m. Just on the basis of thickness differences, comparing growth rates between the two and expecting them to be the same may not be the best course of action. How those rates are calculated in Webber et al. is also important to understand. They took the length of the tourmaline crystals (10 cm) and divided by the cooling time they calculate (5 days). This leads to 10^{-5} cm/s. This is technically a minimum growth rate only based on cooling of the entire pegmatite body, assuming linear growth throughout the entire cooling time of the dike. Therefore, it is feasible these tourmalines grew much faster than what they report as the tourmalines undoubtedly form, grow, and terminate crystallization at different stages during the life time of the pegmatite. This is what we are saying occurred in the Stewart pegmatite.

Otherwise, our rates would be significantly longer.

- “The partition coefficients are treated as constant even though they are known to strongly depend on growth rate (e.g., Huang and Audetat, 2012 for Ti in quartz) as well as the presence of other impurities (e.g., Efremova et al., 2004 for multiple elements in KDP crystals). This could have a significant impact on the inferred growth rates and conclusions. How much wiggle room is there? If one of the K_d 's were off by an order of magnitude, or if the K_d 's were coupled, how would that impact the results?”

The equilibrium partition coefficient has not been shown to change with growth rate except in extreme growth rate cases (see Chpt. 10 in Glicksman, 2010). The effective partition coefficient, however, does, which is what is being measured in Huang & Audetat and modeled here. This is outlined in our theory discussion. Just to reiterate, increased growth rate causes impurities like Ti to pile up at the surface of the crystal. The equilibrium partitioning stays the same, so the crystal ends up taking in more impurities because it only sees the increased concentration at the surface. However, the effective partition coefficient changes, since it is $C_s/C_{\text{far-field}}$. This change is key to our analysis. If the equilibrium k fluctuates with growth rate, both Al and Ge are so highly incompatible and compatible, respectively, that barring a truly drastic change (orders of magnitude), the calculated concentration profiles should not change significantly.

It is true that other impurities can change the partitioning behavior and also may even affect the growth rate (impurity poisoning of the growth surface). This would be an interesting pathway to pursue in the future.

- “In the limit that the boundary layer is infinitely thin, the growth is still rate-limited by nutrient attachment kinetics. In an aqueous solution, for example, I imagine that the Si^{4+} ions (or compounds?) would have a hydration sphere that needs to be shed prior to incorporation into the crystal. I encourage the authors to look into this and check to make sure that the linear growth rates of 1-10 m/day can be accommodated by ion desolvation rates.”

This is an intriguing problem we had not considered. While we have not found any research that has looked specifically at Si^{4+} (or its compounds) desolvation rates, there are rates for other ions in aqueous solution. Looking at Hoffmann et al. (2012), they calculate a water exchange rate of $\sim 10^{10} \text{ s}^{-1}$ for Ca^{2+} , which translates to a desolvation timescale of 10^{-10} s . Our fastest timescale (see Fig. 8) is 240 s, which is significantly slower than the desolvation timescale. While Si^{4+} desolvation may be slower, it will still most likely be fast, compared to our growth timescales.

- “The extreme growth rates imply high supersaturation. If so, what prevents homogeneous or heterogeneous nucleation of many crystals? (which would bring the supersaturation down)”

We do see evidence of nucleation of various phases throughout the chimneys. Because the diffusivities are particularly high, the crystals can grow even while nucleation is occurring. Also, there is evidence of other phases nucleating on quartz as well as quartz nucleating and growing on quartz. We imagine this turbulent system has some complex growth and nucleation dynamics

occurring. The supersaturation may fall quickly, but this should be seen in the trace element profiles. In fact, falling supersaturation may be causing the outer purple rim to appear as a slower growth rate.

REVIEWER COMMENTS

Reviewer #1 (Remarks to the Author):

The authors tried to address my comments and although in some cases they were a bit elusive I'm satisfied by the new version of the manuscript.

I still maintain my doubts about the real stage of crystallisation of the studied quartz samples. Many pegmatites record late hydrothermal crystal growth. For me it is different to document << ultra-fast crystal growth "in" pegmatites >> with respect to << ultra-fast crystal growth "of" pegmatites >>.

It is just a question of a tiny preposition but the meaning is totally different. I understand that choosing "in" the authors were right maintaining a general and open meaning and not implying "that these crystal growths describe the lifetime of a pegmatite or even its evolution".

In this period of "globish" and "fast science", there could be a risk that the lexical difference (in vs. of) in the title will not be understood and that therefore all pegmatites or the entire pegmatite body will form at ultra-fast crystal growth.

For this reason I would add some adjective as "late" or "final" in the title, in abstract and also in the text.

I really believe that pegmatites grow fast but we need data. This manuscript provides some important data about the late stage of crystallisation of a LCT pegmatite body.

all the best
Andrea Dini

Reviewer #2 (Remarks to the Author):

I am satisfied with the adjustments that the authors have made in light of reviewer comments.

Reviewer #3 (Remarks to the Author):

At the end of the day, the conclusion that the growth rates are extremely fast depends primarily on (1) the diffusivities of Ge and Li and (2) built-in model assumptions. The authors showed in the response document that the conclusions are not particularly sensitive to large changes in D (or, for that matter, large changes or uncertainties in temperature). That leaves the model framework and assumptions as warranting careful consideration. With that in mind, my remaining questions/comments are (in order of importance):

(1) The authors infer $R \cdot d \sim 10^{-12} \text{ s}^{-1}$ from the steady state equation (Eq. 4) applied to both Ge and Al (Figure 6). Then, they infer the absolute value of R to be about 10^{-7} m/s from an equation (Supp. Eq. 1 from Smith et al., 1954) that describes the time dependent return to steady state upon a perturbation in the situation where the fluid is static. Comparing the two outcomes, they calculate that d must be about 10^{-5} m (10 micron thin boundary layer) suggesting that the fluid is turbulent. Please explain why this doesn't negate the validity of using Supp. Eq. 1 to produce the model curves in Fig. 7 in the first place.

(2) The authors misunderstood my point about partition coefficients, and this is probably my fault for not being clear enough. It is true that the equilibrium partition coefficient doesn't depend on growth rate because the equations of thermodynamics don't have a time-dependence. The point I was making, however, is that even in situations where diffusion can be assumed to be infinitely fast (no chemical boundary layer), the K_d is known to vary with growth rate – this could be called a “kinetic partition coefficient” or “effective partition coefficient.” In any case, it is different from the effective partition coefficient described in the Response document, which assumes local equilibrium but the concentration of the liquid at the interface differs from that of the far-field. The thing that the authors should be aware of (and acknowledge in the paper) is that the local equilibrium assumption may not be valid (could cite DePaolo, 2011, for example), and if it isn't, then the conclusions could change. At this juncture there's not enough information in the literature to re-build the model with a rate-dependent K_d and so I think that what the authors have done is okay but it would be a good idea for them to hedge their bet and point out that this specific assumption may need to be relaxed in the future.

(3) This is an easy one to fix. I made a mistake in my review when talking about the relevant pressure ranges of the TitaniQ calibrations. The numbers I quoted were off by one order of magnitude. That aside, the authors should use and cite the Thomas et al. (2010) calibration for estimating T (5-20 kbar) because the Wark and Watson (2006) calibration only applies to $P = 10$ kbar – it is not safe to extrapolate the Wark and Watson (2006) calibration to the relevant pressures (1.5 to 2 kbar) because Ti-in-quartz is strongly pressure-dependent.

(4) This is probably easy to address. The paper does not speculate on what caused the inferred three order of magnitude increase in growth rate. In the response doc, they state that they think that the opening of chimneys caused a pressure drop that induced quartz crystallization. Why isn't this pressure drop recorded in the Ti profile (Figure 2)? Is it because the Ti values are near detection limits and so it's really just a profile of “noise”?

(5) This is a simple question that the future reader may also ask. On line 161, it says “incompatible elements in the growing crystal will first increase before falling back to a steady state.” With Al being incompatible ($k = 0.02$; line 274) why does the model Al profile increase monotonically in Fig. 7?

(6) On line 65 of the Supplement, Smith et al. (1953) should be (1955).

We thank the reviewers for their critical feedback. The manuscript has been significantly improved because of it. Below, we respond to specific comments. We also provide a track changes document showing where in the text we revised.

Response to Reviewer #1's comments:

- The authors tried to address my comments and although in some cases they were a bit elusive I'm satisfied by the new version of the manuscript. I still maintain my doubts about the real stage of crystallisation of the studied quartz samples. Many pegmatites record late hydrothermal crystal growth. For me it is different to document << ultra-fast crystal growth "in" pegmatites >> with respect to << ultra-fast crystal growth "of" pegmatites >>. It is just a question of a tiny preposition but the meaning is totally different. I understand that choosing "in" the authors were right maintaining a general and open meaning and not implying "that these crystal growths describe the lifetime of a pegmatite or even its evolution". In this period of "globish" and "fast science", there could be a risk that the lexical difference (in vs. of) in the title will not be understood and that therefore all pegmatites or the entire pegmatite body will form at ultra-fast crystal growth. For this reason I would add some adjective as "late" or "final" in the title, in abstract and also in the text.

We thank the reviewer for this comment. We have clarified in the discussion section that our crystallization rates only apply to the miarolitic stage of pegmatite evolution, and it does not necessarily relate to any other part of the pegmatite crystallization process. However, we do not think putting "late" into the title is appropriate. Instead, we have changed the title to "Episodes of fast crystal growth..." to imply we are seeing one snapshot of crystal growth during a pegmatite's evolution.

Response to Reviewer #3's comments:

- The authors infer $R \cdot d \sim 10^{-12} \text{ s}^{-1}$ from the steady state equation (Eq. 4) applied to both Ge and Al (Figure 6). Then, they infer the absolute value of R to be about 10^{-7} m/s from an equation (Supp. Eq. 1 from Smith et al., 1954) that describes the time dependent return to steady state upon a perturbation in the situation where the fluid is static. Comparing the two outcomes, they calculate that d must be about 10^{-5} m (10 micron thin boundary layer) suggesting that the fluid is turbulent. Please explain why this doesn't negate the validity of using Supp. Eq. 1 to produce the model curves in Fig. 7 in the first place.

The reviewer's comments are relevant. We have looked more closely at our assumptions and have clarified in the text. The first issue is whether the steady state equation (Eq. 4) is relevant for dynamic systems. This application to dynamic systems is fine because chemical transport to the growing crystal *within* the boundary layer is controlled only by diffusion. This is analogous to thermal convection, wherein transport of heat in the thermal boundary layer is controlled by diffusion and can thus be approximated by static thermal diffusion, but the thickness of the thermal boundary layer is controlled by advection after steady state is achieved. So Eq. 4 is valid

for the core region. The second issue is whether Supp Eq. 1, which is used to model the transient response in the orange region, is valid for dynamic systems. The reviewer is right that it is not perfectly valid, and we have noted this now. However, it does provide a minimum bound on growth rates. R scales with D/d . Assuming D is constant, then advection will decrease d , resulting in an increase in R . So our estimates of R from Supp. Eq. 1 are minimum bounds and our d s are maximum bounds in an advecting system

- The authors misunderstood my point about partition coefficients, and this is probably my fault for not being clear enough. It is true that the equilibrium partition coefficient doesn't depend on growth rate because the equations of thermodynamics don't have a time-dependence. The point I was making, however, is that even in situations where diffusion can be assumed to be infinitely fast (no chemical boundary layer), the K_d is known to vary with growth rate – this could be called a “kinetic partition coefficient” or “effective partition coefficient.” In any case, it is different from the effective partition coefficient described in the Response document, which assumes local equilibrium but the concentration of the liquid at the interface differs from that of the far-field. The thing that the authors should be aware of (and acknowledge in the paper) is that the local equilibrium assumption may not be valid (could cite DePaolo, 2011, for example), and if it isn't, then the conclusions could change. At this juncture there's not enough information in the literature to re-build the model with a rate-dependent K_d and so I think that what the authors have done is okay but it would be a good idea for them to hedge their bet and point out that this specific assumption may need to be relaxed in the future.

The reviewer makes a good point here, and we have adjusted the manuscript accordingly, including citing DePaolo, 2011 (line 229-233). We agree with the reviewers that there is insufficient knowledge now to account for these effects.

- This is an easy one to fix. I made a mistake in my review when talking about the relevant pressure ranges of the TitaniQ calibrations. The numbers I quoted were off by one order of magnitude. That aside, the authors should use and cite the Thomas et al. (2010) calibration for estimating T (5-20 kbar) because the Wark and Watson (2006) calibration only applies to $P = 10$ kbar – it is not safe to extrapolate the Wark and Watson (2006) calibration to the relevant pressures (1.5 to 2 kbar) because Ti-in-quartz is strongly pressure-dependent.

We have included this change along with the calibration from Huang & Audetat (2012).

- This is probably easy to address. The paper does not speculate on what caused the inferred three order of magnitude increase in growth rate. In the response doc, they state that they think that the opening of chimneys caused a pressure drop that induced quartz crystallization. Why isn't this pressure drop recorded in the Ti profile (Figure 2)? Is it because the Ti values are near detection limits and so it's really just a profile of “noise”?

The Ti in Figure 2 is noisy and near the detection limit. A better idea of the Ti change can be seen in Supplementary Figure 1, which does show an increase, along with the other incompatible elements.

- This is a simple question that the future reader may also ask. On line 161, it says “incompatible elements in the growing crystal will first increase before falling back to a steady state.” With Al being incompatible ($k = 0.02$; line 274) why does the model Al profile increase monotonically in Fig. 7?

There is a relatively simple explanation. The crystal has not grown long enough to reach a peak in concentration, so it simply increases monotonically during this early transient state.

- On line 65 of the Supplement, Smith et al. (1953) should be (1955).

Thank you for catching this. It has been fixed.

REVIEWERS' COMMENTS:

Reviewer #3 (Remarks to the Author):

In this 3rd round of review, the authors have addressed all of my previous comments to my satisfaction.